# Prevalence of Monosodium Urate (MSU) Deposits in Cadavers Detected by Dual-Energy Computed Tomography (DECT)

**DOI:** 10.3390/diagnostics12051240

**Published:** 2022-05-16

**Authors:** Andrea S. Klauser, Sylvia Strobl, Christoph Schwabl, Werner Klotz, Gudrun Feuchtner, Bernhard Moriggl, Julia Held, Mihra Taljanovic, Jennifer S. Weaver, Monique Reijnierse, Elke R. Gizewski, Hannes Stofferin

**Affiliations:** 1Department of Radiology, Medical University Innsbruck, 6020 Innsbruck, Austria; andrea.klauser@i-med.ac.at (A.S.K.); gudrun.feuchtner@i-med.ac.at (G.F.); elke.gizewski@i-med.ac.at (E.R.G.); 2Department of Internal Medicine II, Medical University Innsbruck, 6020 Innsbruck, Austria; werner.klotz@tirol-kliniken.at (W.K.); julia.held@tirol-kliniken.at (J.H.); 3Department of Anatomy, Histology and Embryology, Institute of Clinical and Functional Anatomy, Medical University Innsbruck, 6020 Innsbruck, Austria; bernhard.moriggl@i-med.ac.at (B.M.); hannes.stofferin@i-med.ac.at (H.S.); 4Department of Medical Imaging, Banner University Medical Center, College of Medicine, The University of Arizona, Tucson, AZ 85724, USA; mihrat@radiology.arizona.edu; 5Department of Radiology, University of New Mexico, Albuquerque, NM 87131, USA; jsweaver@salud.unm.edu; 6Division of Musculoskeletal Radiology, Department of Radiology, Leiden University Medical Center, 2333 ZC Leiden, The Netherlands; m.reijnierse@lumc.nl

**Keywords:** gout, monosodium urate deposits, cardiovascular, musculoskeletal, dual-energy computed tomography, ocular lense, tendons, kidney

## Abstract

Background: Dual-energy computed tomography (DECT) allows direct visualization of monosodium urate (MSU) deposits in joints and soft tissues. Purpose: To describe the distribution of MSU deposits in cadavers using DECT in the head, body trunk, and feet. Materials and Methods: A total of 49 cadavers (41 embalmed and 8 fresh cadavers; 20 male, 29 female; mean age, 79.5 years; SD ± 11.3; range 52–95) of unknown clinical history underwent DECT to assess MSU deposits in the head, body trunk, and feet. Lens, thoracic aorta, and foot tendon dissections of fresh cadavers were used to verify MSU deposits by polarizing light microscopy. Results: 33/41 embalmed cadavers (80.5%) showed MSU deposits within the thoracic aorta. 11/41 cadavers (26.8%) showed MSU deposits within the metatarsophalangeal (MTP) joints and 46.3% of cadavers demonstrated MSU deposits within foot tendons, larger than and equal to 5 mm. No MSU deposits were detected in the cranium/intracerebral vessels, or the coronary arteries. Microscopy used as a gold standard could verify the presence of MSU deposits within the lens, thoracic aorta, or foot tendons in eight fresh cadavers. Conclusions: Microscopy confirmed the presence of MSU deposits in fresh cadavers within the lens, thoracic aorta, and foot tendons, whereas no MSU deposits could be detected in cranium/intracerebral vessels or coronary arteries. DECT may offer great potential as a screening tool to detect MSU deposits and measure the total uric acid burden in the body. The clinical impact of this cadaver study in terms of assessment of MSU burden should be further proven.

## 1. Introduction

Gout is a crystal-induced inflammatory arthritis with increasing incidence and prevalence in recent decades [1]. It represents a major healthcare burden, given its association with metabolic syndrome, coronary heart disease, and diabetes mellitus [1]. 

Dual-energy computed tomography (DECT) is a well-established method for the detection of MSU deposits in peripheral joints and tendons and has been implemented in the American College of Radiology/European League Against Rheumatism (ACR/EULAR) 2018 gout classification criteria [2,3,4,5]. 

Chhana et al. [6] described DECT as an advanced imaging method for the assessment of crystal proven tophaceous gout in 12 different joints but only in one cadaveric specimen. A recent study by Klauser et al. [7] demonstrated the usefulness of DECT for the detection of cardiovascular MSU deposits in patients with gout and a control group of patients without a previous history of gout or inflammatory rheumatic disease, as verified by microscopy in fresh cadavers. Especially in the last few years, studies concerning extraarticular gout manifestations have become more and more frequent [8,9]. Nevertheless, there is still a broad discussion regarding subclinical or vascular deposits [10,11,12]. A general consensus regarding the optimal DECT protocol has not yet been established. it remains questionable how many of the MSU deposits found in the DECT are actual urate or merely artifacts [13,14,15,16].

To our knowledge, direct imaging of MSU deposits of the head, body trunk, and feet in embalmed cadavers by DECT has not been reported to date.

The detection of artefacts in DECT is a frequent point of discussion and several typical artefacts have been well described in the literature, e.g., finger nails [17,18]. Due to the lack of verification by microscopy, we do not yet know, if the MSU deposits are always true or artefacts. To exclude artefacts and verify the MSU deposits found with DECT, crystal characterization with polarizing light microscopy was performed in the lens, aorta, and feet tendons in fresh cadavers.

## 2. Materials and Methods

### 2.1. Cadavers

A total of 49 cadavers, 41 embalmed (16 male, 25 female, mean age, 82 years; SD ± 16.4; range 52–91) and 8 fresh (4 male, 4 female, mean, 75 years; SD ± 13.2; range 72–95) were enrolled from 1 January 2017, and through 1 November 2018. 

Informed consent was provided according to the last wills of the donors, who had donated their bodies to human research studies. Institutional review board approval was obtained. All embalmed and fresh cadavers were referred to DECT after death and were in legal custody of the Anatomy institution. 

No medical history was available including gouty arthritis or hyperuricemia. DECT examinations of the head, body trunk, and feet were performed.

### 2.2. DECT Scan Parameters

The evaluation was performed with a 128-row dual-source CT scanner (Somatom Definition Flash; Siemens Healthineers, Forchheim, Germany) at two energy levels (80 and 140 kV) using two separate sets of X-ray tubes and detectors positioned 90 to 95 degrees apart without the use of contrast media. The standardized protocol settings included 80 kV/100–140 mAs for tube A and 140 kV/200–250 mAs for tube B, with a ratio of 1.36, range of 4, minimum HU of 150, and maximum HU of 500. Scan parameters were 2 × 64 × 0.625 mm acquisition, rotation time of 1 s, DLP 219 mGycm, CTDI vol 11.01 L, total mAs 3415, slice thickness of 0.75 mm, and increment of 0.5 mm. Axial, coronal, and sagittal reformations were reconstructed from the DE datasets at a resolution of 0.4 mm with soft tissue kernel (D30) and bone kernel (B60). D30 kernel was used for DE processing and MSU detection.

The acquired datasets were reconstructed in the desired planes and processed with dual-energy software utilizing a standardized two-material decomposition algorithm designed for specific clinical applications [19]. The two-material decomposition algorithm is based on the principle that materials with a high atomic number such as calcium would demonstrate a higher increase in attenuation at higher photon energies than does a material composed of low atomic number materials such as MSU crystals. Once separated and characterized, the materials were color-coded and overlaid on multi-planar reformatted cross-sectional images [19]. We choose green pixels for MSU deposit demonstration when using the software of the Syngovia workstation (Siemens Healthineers). 

Pre-processed and processed images were transferred to the picture archiving system (PACS). Corresponding pre-processed grey-scale images are reviewed for the presence of deposits (8).

### 2.3. CT and DECT Scoring

Two radiologists with experience in gout imaging by DECT of 5 and 7 years evaluated the DECT images in consensus.

Anatomic locations of calcified plaques and MSU deposits were determined as follows:+ Head/Neck:
Cranium/intracerebral vessels, ear cartilage and orbits (lens)Supraaortal vessels (Subclavian and Carotid arteries)
+ Body trunk
Ascending aorta, descending aorta, aortic arch, aortic root, abdominal aortaRight coronary artery (RCA), left main artery (LM), circumflex artery (CX) and left anterior descending artery (LAD)Tricuspid valve and mitral valveIliac vesselsRib cartilagesKidney


Cardiovascular calcified plaques with a CT attenuation >130 Houndsfield Units (HU) [20] and MSU deposits in the thoracic aorta, coronary arteries, valves, abdominal aorta, and iliac vessels were graded according to Gondrie et al. [21] as follows:

Score 0 = absent, score 1 = 5 or fewer foci, score 2 = between 6–8 foci and extending over 3 section, score 3 = more than 9 foci extending over 3 sections.

+ Feet:
Joints: Metatarsophalangeal (MTP) joints, interphalangeal joints (IP), tibiotalar joint.Tendons: Extensor hallucis longus tendon (EHL), tibialis anterior tendon (TAT), tibialis posterior tendon (TPT), flexor hallucis longus tendon (FHL), peroneal tendons (PT), and Achilles tendon (AT).


MSU deposits in MTP joints and foot tendons were graded as follows:

Score 0 = absent, Score 1 = MSU deposits < 5 mm, Score 2 = MSU deposits between 5–10 mm, Score 3 = MSU deposits ≥ 10 mm.

According to ACR/EULAR guidelines nail bed deposits, submillimeter deposits, skin deposits, and deposits by beam hardening and vascular artefacts were not classified as positive findings in our study [3].

### 2.4. Polarizing Microscopic Evaluation

In 8 fresh cadavers with DECT positive MSU deposits within the lens, thoracic aorta, or foot tendons, gross anatomical sectioning according to defined landmark was performed, cut unfixed to pieces of 5 mm × 5 mm, embedded using Tissue-Tek^®®^ O.C.T.™ compound medium and sectioned at 5 μm using a Leica CM1950 S cryostat. After mounting on microscope slides and covering using Glycerine/Phosphate-Buffered Saline (PBS) solution, cryostat section examination was performed with compensated polarized light microscopy at 400× magnification. First order red compensation was performed to recognize MSU crystals by their needle-like appearance and strong negative birefringence.

### 2.5. Statistical Analysis

Statistical analysis was performed using R Project for Statistical Computing 3.5.1 [R Core Team (2013). R: A language and environment for statistical computing. R Foundation for Statistical Computing, Vienna, Austria. http://www.R-project.org]. The presence of MSU deposits and calcified plaques for the different anatomical locations was tabulated together with the individual scores. To analyze the relationship between MSU deposits and the occurrence of calcified plaques contingency tables were generated and a Fisher’s Exact Test for count data was performed. To test scoring results for age dependence Spearman’s rank correlation coefficients were calculated. Results were considered significant for *p*-values less than 0.05.

## 3. Results

### 3.1. Fresh Cadavers

1 cadaver demonstrated MSU deposits within the orbits (Figure 1), 1 cadaver showed MSU deposits within the thoracic aorta (Figure 2) and all cadavers showed MSU deposits within the foot tendons (AT, PT, FHL, TAT) (Figure 3). MSU deposits detected by DECT were histologically proven to be present by polarized light microscopy in a total of 10/10 biopsies (100%).

### 3.2. Embalmed Cadavers

In the head, 4/41 (9.8%) cadavers showed MSU deposits within the lens and 40/41 (97.6%) of the cadavers demonstrated MSU deposition within the ear cartilage. No MSU deposits were detected within the neurocranium and intracerebral vessels. 30/41 cadavers (73.2%) showed MSU deposits within the thoracic aorta and all cadavers demonstrated calcified aortic wall plaques (Table 1). Only 1/41 cadavers (2.4%) showed MSU deposits within the abdominal aorta.

For the ascending aorta, Fisher’s Exact Test showed a significant association between MSU deposits and calcified plaques (*p* = 0.02). Spearman’s rank correlation showed a weak but significant positive dependence on age for calcified plaques (rho = 0.332, *p* = 0.034); however, no dependence on age was demonstrated for MSU deposits (rho = −0.07, *p* = 0.68). All other vessels or regions in the body trunk demonstrated no significant association or age dependence for calcified plaques or MSU deposits.

None of the cadavers were positive for MSU deposits in the coronary arteries and valves (Table 1). Only calcified plaques within the LAD showed a weak but significant positive dependence on age (rho = 0.387, *p* = 0.012); all other coronary arteries showed no dependence on age. In all 41 cadavers (100%) MSU deposits were detected in costochondral cartilages (Figure 2 and Figure 4), and 3/41 (7.3%) showed MSU deposits within kidney stones (Figure 4).

11/41 (26.8%) cadavers showed MSU deposits within the MTP joints (Table 2). The first MTP joint was the most commonly involved joint in 6/11 MTP positive cadavers (54.5%) (Table 2). 

46.3% of cadavers demonstrated MSU deposits within foot tendons, larger than 5 mm (Table 3). The most common Score 3 MSU deposition site was the AT at 31.7%, followed by TAT at 19.5% (Table 3).

## 4. Discussion

DECT screening for gout deposits in cadavers shows a high prevalence. Histological correlation in fresh cadavers could confirm these positive DECT findings as MSU deposits (and not artefacts). MSU deposits are predominantly seen within vessel walls in calcified plaques (however not in the brain), in rib cartilage, and in foot tendons. 

In hyperuricemia, supersaturation leads to the precipitation of urate crystals in the plasma and within joints. Additionally, all other interstitial fluids will be supersaturated as well and may drive crystal-induced inflammation. Tophi can present in unexpected anatomical locations; therefore vigilance is required when unusual symptoms or signs occur in a patient suspected to have gout [22]. 

DECT has emerged as a useful diagnostic imaging modality for the diagnosis of gout, by offering the advantage of directly assessing MSU deposits as well as displaying the anatomic extent of the disease [2,5]. DECT can be used to monitor response to drug therapy and maximize clinical management, thus optimizing patient outcomes [23]. It has been impacted in patients to urate lowering therapy [24].

DECT detects MSU deposits in peripheral joints with sensitivities of 78 to 100% and specificities of 78 to 100% [25,26,27]. MTP 1 joint is the most affected joint in gouty arthritis [28], consistent with our findings in 6/11 (54.5%) MTP positive embalmed cadavers. Pascart et al. [29] showed that the extent of the MSU burden in peripheral joints (knee joint and feet joints) measured with DECT predicted the risk of flares. 

However, in the last 30 years, only one study evaluated cadaveric MTP 1 joints [30] and reported that 12/70 (17.1%) consecutive autopsies showed MSU crystal deposits in polarizing light microscopy. 2/12 (16.7%) subjects had a history of podagra or gouty arthritis pre-mortem in this study. This suggests a possible post-mortem crystallization since the level of serum uric acid necessary for supersaturation decreases with a reduction in body temperature from 6.8 mg% at 37° to 2.4 mg% at 30° as previously reported by Loeb et al. [31]. Post-mortem crystallization could therefore also explain higher numbers of positive MTP 1 joints (6/11 positive MTP joints) in our DECT study.

Tendon involvement in patients with gout is frequent and affects joint stability and flexibility. Racide et al. [32] postulated that tendons rupture at the sites of MSU crystal deposits because the urate crystals lead to a reduction in the tensile strength of the tendon. Tendon rupture due to tophus infiltration has been described in patients with chronic gout, although this is an uncommon event [33]. 

In a study by Dalbeth et al. [34] in 92 patients, MSU deposits in tendons occurred in tophaceous gout. The most commonly involved tendon in the foot was the AT (39.1%), followed by the PT (18.1%). The TAT and the extensor tendons were involved less commonly in 7.6–10.3%. These findings are in line with our study, where the most commonly involved Score 3 MSU deposition site was the AT at 31.7%, followed by TAT at 19.5%.

Yuan et al. [35] also reported tendons as the most frequent anatomical location of MSU deposits, in up to 41.4%, consistent with our findings. It is unknown if small MSU deposits (Score 1 and 2) in tendons are of clinical importance; this should be investigated in future studies. Furthermore, postprocessing seems to take an important role, to increase sensitivity for MSU deposits not only in joints but especially in tendons, as seen in previous studies [16,36], which should also be assessed in further investigations.

The gouty infiltration of tendons and soft tissues should hence also be considered a rare differential diagnosis for nonspecific soft tissue masses, even in asymptomatic patients with unknown hyperuricemia prior to their first tophaceous manifestation [37].

Previous studies have shown a high incidence of uric acid kidney stones with the highest prevalence in the middle east and in Europe [38]. DECT is useful in discriminating uric acid stones and other stone types with 92–100% accuracy, a positive predictive value of 100%, and a negative predictive value of 98.5% [39]. In a recent study by Li et al. [40] nephrolithiasis was reported in 27/84 (32.1%) patients with gout, with a high incidence of pure uric acid (UA) kidney stones in 17/27 (63%). DECT imaging may permit patients with UA stones to benefit from conservative treatment and avoid interventional procedures. We detected a low number of 3/41 (7.3%) UA stones in our cadaveric study.

Confirmed urate crystals in the eyes have been rarely reported in the past 40 years [41]. The majority of gout patients’ ocular abnormalities are asymptomatic [42]. Precipitation of urate crystal has been described in eyelids, tarsal plates, conjunctiva, cornea, sclera, tendons of extraocular muscles, orbit, and lens. It has been reported that tophi can be deposited in the iris or in the anterior chamber, causing anterior uveitis or glaucoma [43]. Lin et al. [42], found uric acid crystals on the ocular surface in 3/380 (0.79%) consecutive gout patients. They further reported that patients with UA crystals in their ocular surface also showed many tophi in both ears, finger roots, feet, and kidneys, but this was not confirmed by DECT. In our study 1/4 lens positive embalmed cadavers also showed UA kidney stones, 2/4 MSU deposits in MTP joints, and all of them (4/4) showed MSU deposits in rib cartilages and were positive for MSU deposits in foot tendons. A possible correlation should be investigated in further studies.

In general, we demonstrated MSU deposits in the lens in 4/41 embalmed cadavers (9.8%), which was proven by polarizing light microscopy in fresh cadavers. Ferry et al. [44] reported in 69 severe gout patients that the most common abnormality was bilateral ocular redness caused by hyperemia of conjunctival and episcleral vessels. When evaluating a patient with bilateral chronic conjunctival redness, the clinician should consider MSU deposits in the differential diagnosis. A statistically significant difference in subconjunctival hemorrhage in gout patients and other groups was reported, whereas the subconjunctival hemorrhage in gout patients was not absorbed after three months. This underscores the importance of early detection of ocular MSU deposits [42].

Only one DECT study by Carr et al. [45] evaluated extensive MSU deposits in rib cartilage, in 19/20 gout patients (95%) without any difference between patients and controls indicating that it is not a disease-specific finding but instead represents a physiologic process. This is consistent with our findings, where all cadavers (100%) were positive for MSU deposits in rib cartilages. Further studies should prove whether physiologic MSU deposits in rib cartilage occur. MSU deposits in ear cartilage have only been scarcely reported in a few case reports. We found a high frequency of MSU deposits in ear cartilage in 40/41 cadavers (97.6%). Tophi in the external middle ear or ear helix should be considered in the differential diagnosis of ear masses alongside common pathologies [46]. 

Kumral et al. demonstrated that high uric acid levels are strongly associated with CAD and that elevated uric acid might be injurious for large cerebral arteries [47]. To our knowledge, this is the first study performing DECT imaging of the neurocranium and intracerebral vessels in cadavers. Interestingly MSU deposits in intracerebral vessels could not be detected in our study and also have not been reported to date. Karagiannis et al. [48] showed that serum UA is an independent predictor for early death after acute stroke and UA lowering therapy improved outcomes in stroke patients receiving intravenous thrombolysis followed by thrombectomy. McFarland et al. [49] reported an inverse association between urate levels and Parkinson Disease, Lewy body dementia, and possibly Alzheimer’s disease.

Among patients with gout, characteristic gout severity factors are associated with cardiovascular disease (CVD). Disveld et al. [50] showed that crystal-proven gout is strongly associated with an increased prevalence of CVD. This underscores the importance of early detection of cardiovascular MSU deposits. Andres et al. [51] reported a significant increase in coronary calcification and MSU deposits in the knees and MTP 1 joints in patients with asymptomatic hyperuricemia. However, these calcifications were not evaluated in terms of cardiovascular MSU deposits, as assessed in our study. In contrast, Pascart et al. [52] found that the extent of MSU burden in knees and feet detected by DECT and ultrasound did not increase the estimated risk of a cardiovascular event in 42 gout patients. 

Only a paucity of gouty tophi in the mitral valve, aortic valve, and pulmonal valve has been previously reported [53]. In our study, mitral and tricuspid valves were evaluated and did not show any MSU deposits. Our preliminary findings can support the hypothesis of an association between calcified plaques and cardiovascular deposition of MSU in the ascending aorta in 30/41 cadavers (73.2%), but not in coronary arteries, supraaortal vessels, and valves, which has recently been described in assessing gout patients using DECT [7,54]. Several studies reflect the suboptimal care received by gout patients and suggest an urgent need to optimize treatment and prevent adverse outcomes. As previously reported, DECT provides the option of quantifying the MSU burden [55] and hence can be used to monitor the response of patients during MSU lowering therapy [56]. In future studies, it may be interesting to characterize and volumetrize the deposits, especially in the vessel walls. Artificial intelligence sofware could also be used to facilitate the analysis, which has already been successfully applied in other regions [57,58,59].

### Limitations

Prior medical history of the cadavers was not available. 

Volume rendering imaging of MSU deposits was not performed and should be investigated in further clinical studies to quantify gouty burden.

This study is limited by the small sample size for cardiovascular studies and therefore larger cohorts are stringent necessary in order to define the prognostic value of MSU deposits for CVD outcomes.

## 5. Conclusions

DECT can be used to detect MSU deposits in both fresh and embalmed cadavers. DECT showed MSU deposits within the lens, thoracic aorta, and foot tendons of our cadavers, whereas no MSU deposits could be detected in neurocranium/intracerebral vessels and coronary arteries. Findings were confirmed with microscopy. In addition to detecting MSU deposits within the peripheral joints and tendons, DECT offers the potential to image MSU deposits in the other body parts and organ systems and thus may be feasible as a screening tool to detect MSU deposits and to measure total uric acid burden in the whole body. 

The clinical impact of this cadaver study should be further investigated.

## Figures and Tables

**Figure 1 diagnostics-12-01240-f001:**
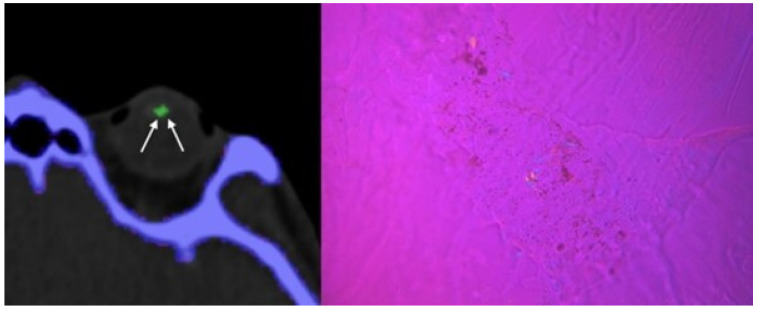
80-year-old male cadaver. Axial DECT image showing Score 1 MSU deposit in the left lens (white arrows). Corresponding microscopic image taken from the lens, showing diffuse packed and patchy MSU deposits with strong negative birefringence (bluish structures).

**Figure 2 diagnostics-12-01240-f002:**
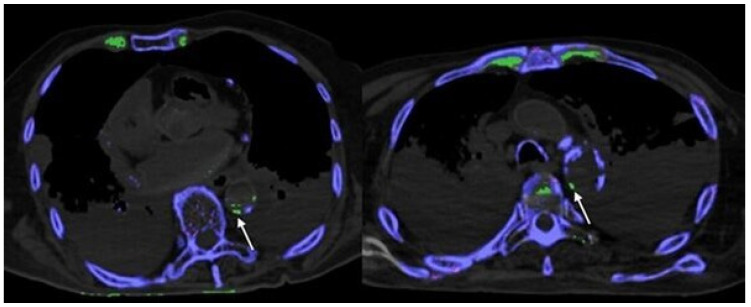
Axial DECT images showing histologically verified Score 1 MSU deposits in thoracic aorta (white arrows). Note: MSU deposits in anterior costochondral cartilage and intervertebral disc.

**Figure 3 diagnostics-12-01240-f003:**
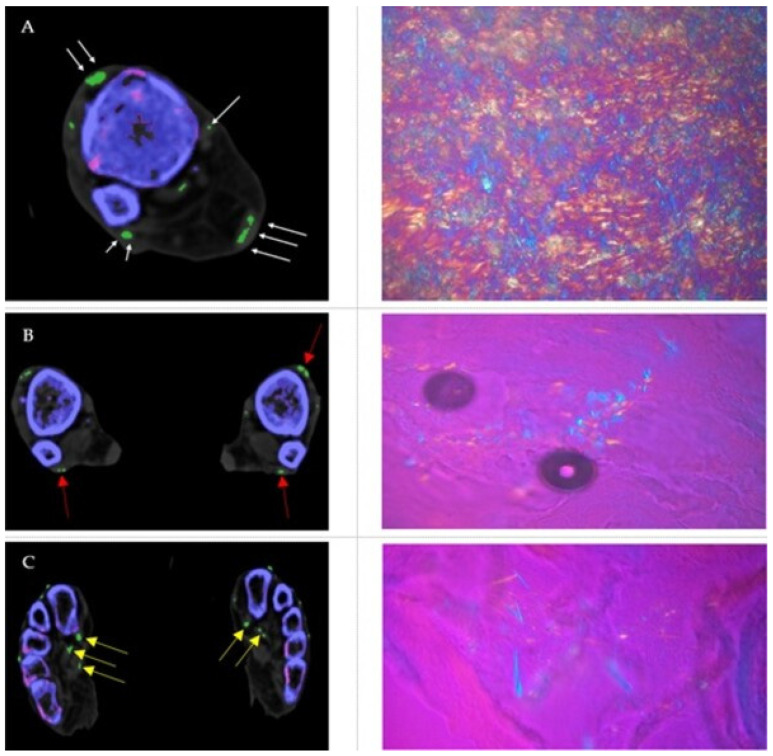
(**A**) Axial DECT image showing MSU deposits within the TAT, PT, AT, and TPT (white arrows) with corresponding microscopic image taken from the AT, showing large diffuse packed and patchy MSU crystals with strong negative birefringence (bluish structures). (**B**) Axial DECT image demonstrating MSU deposits within the TAT and PT (red arrows) with corresponding microscopic image taken from the PT. (**C**) Axial DECT image showing MSU deposits within the FHL and flexor digitorum longum tendon (yellow arrows) with corresponding microscopic image taken from the FHL.

**Figure 4 diagnostics-12-01240-f004:**
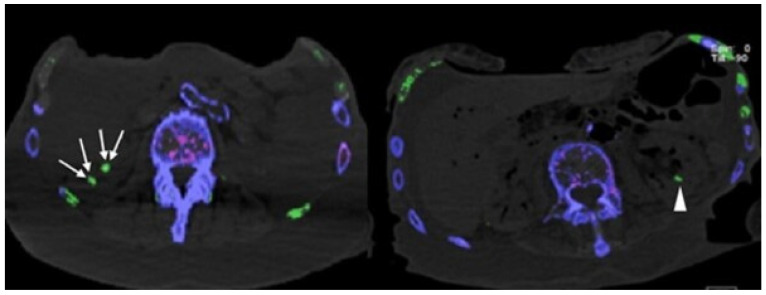
Axial DECT images showing MSU deposits in kidney stones (white arrows) and MSU deposit in renal calyx (arrowhead).

**Table 1 diagnostics-12-01240-t001:** Localization and scoring of calcified plaque (first line) and MSU deposits (second line) in embalmed cadavers.

Anatomical Location	Score 0	Score 1	Score 2	Score 3
Aorta				
root	6/41 (14.6%) 31/41 (75.6%)	0/41 (0%) 10/41 (24%)	11/41 (26.9%) 0/41 (0%)	24/41 (58.5%) 0/41 (0%)
ascending	13/41 (31.7%) 34/41 (82.9%)	0/41 (0%) 7/41 (17.1%)	7/41 (17.1%) 0/41 (0%)	21/41 (51.2%) 0/41 (0%)
arch	1/41 (2.4%) 31/41 (75.6%)	0/41 (0%) 10/41 (24%)	3/41 (7.3%) 0/41 (0%)	37/41 (90.2%) 0/41 (0%)
descending	2/41 (4.9%) 13/41 (31.7%)	0/41 (0%) 22/41 (53.7%)	3/41 (7.3%) 2/41 (4.9%)	36/41 (87.9%) 4/41 (9.8%)
abdominal	0/41 (0%) 40/41 (97.6%)	2/41 (4.9%) 1/41 (2.4%)	0/41 (0%) 0/41 (0%)	39/41 (95.1%) 0/41 (0%)
Supraaortal vessels	2/41 (4.9%) 28/41 (68.3%)	0/41 (0%) 12/41 (29.3%)	6/41 (14.6%) 1/41 (2.4%)	33/41 (80.5%) 0/41 (0%)
Iliac vessels	0/41 (0%) 36/41 (87.8%)	2/41 (4.9%) 5/41 (12.2%)	0/41 (0%) 0/41 (0%)	39/41 (95.1) 0/41 (0%)
Valves				
tricuspid	41/41 (100%) 41/41 (100%)	0/41 (0%) 0/41 (0%)	0/41 (0%) 0/41 (0%)	0/41 (0%) 0/41 (0%)
mitral	41/41 (100%) 41/41 (100%)	0/41 (0%) 0/41 (0%)	0/41 (0%) 0/41 (0%)	0/41 (0%) 0/41 (0%)
Coronary arteries				
LAD	2/41 (4.9%) 41/41 (100%)	0/41 (0% 0/41 (0%)	3/41 (7.3%) 0/41 (0%)	36/41 (87.9%) 0/41 (0%)
LM	11/41 (26.8%) 41/41 (100%)	0/41 (0% 0/41 (0%)	3/41 (7.3%) 0/41 (0%)	27/41 (65.9%) 0/41 (0%)
RCA	8/41 (19.5%) 41/41 (100%)	0/41 (0% 0/41 (0%)	5/41 (12.2%) 0/41 (0%)	28/41 (68.3%) 0/41 (0%)
CX	8/41 (19.5%) 41/41 (100%)	0/41 (0% 0/41 (0%)	4/41 (9.8%) 0/41 (0%)	29/41 (70.7%) 0/41 (0%)

Note: Score 0 = absent, score 1 = 5 or fewer foci, score 2 = 6–8 foci, >3 section, score 3 ≥9 foci, >3 sections. LAD = left anterior descending artery, LM = left main artery, RCA = right coronary artery, CX = circumflex artery.

**Table 2 diagnostics-12-01240-t002:** Scoring of MSU deposits in foot joints of embalmed cadavers.

Anatomical Location	Positive Joints	Score 0	Score 1	Score 2	Score 3
MTP joints	11/41 (26.8%)	30/41 (73.2%)	2/41 (4.9%)	6/41 (14.6%)	3/41 (7.3%)
MTP I	6/11 (54.5%)		1/6 (16.7%)	3/6 (50%)	2/6 (33.3%)
MTP I − V	4/11 (36.4%)		0/4 (0%)	3/4 (75%)	1/4 (25%)
MTP I + V	1/11 (10%)		1/1 (100%)	0/1 (0%)	0/1 (0%)
Interphalangeal joint	0/41 (0%)				
Ankle	0/41 (0%)				
Tarsus	0/41 (0%)				

Note: Score 0 = absent, score 1 = 5 or fewer foci, score 2 = 6–8 foci, >3 section, score 3 ≥ 9 foci, >3 sections. MTP = metatarsophalangeal.

**Table 3 diagnostics-12-01240-t003:** Scoring of MSU deposits in foot tendons of embalmed cadavers.

Anatomical Location	Score 0	Score 1	Score 2	Score 3
EHL	1/41 (2.4%)	24/41 (58.5%)	11/41 (27.5%)	5/41 (12.2)
FHL	1/41 (2.4%)	31/41 (75.6%)	8/41 (19.5%)	1/41 (2.4%)
TAT	1/41 (2.4%)	13/41 (31.7%)	19/41 (46.3%)	8/41 (19.5%)
TPT	1/41 (2.4%)	31/41 (75.6)	8/41 (19.5%)	1/41 (2.4%)
PT	1/41 (2.4%)	19/41 (46.3%)	18/41 (43.9%)	3/41 (7.3%)
AT	1/41 (2.4%)	22/41 (53.6%)	5/41 (12.2.%)	13/41 (31.7%)

Note: Score 0 = absent, Score 1 = MSU deposits < 5 mm, Score 2 = MSU deposits between 5–10 mm, Score 3 = MSU deposits ≥ 10 mm. EHL = extensor hallucis longus, FHL = flexor hallucis longus, TAT = tibialis anterior tendon, TPT = tibialis posterior tendon, PT = peroneal tendons, AT = achilles tendon.

## Data Availability

Data supporting reported results may be provided upon reasonable request.

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
