# Peer review of "Prevalence of Monosodium Urate (MSU) Deposits in Cadavers Detected by Dual-Energy Computed Tomography (DECT)"

_diagnostics, 2022, doi:10.3390/diagnostics12051240_

Round 1

Reviewer 1 Report

Enjoyed reading this article. Very interesting data. The text in general needs minor edits with spelling and punctuation errors. There are whole paragraphs consisting of one sentence each. It can also be made more concise. Some sentences are redundant.  

Line 26- should this be greater than and equal to 5mm per classification later on in text?

Line 54- "To exclude artefacts, MSU crystal characterization with polarizing light microscopy was performed using the lens, aorta and feet tendons in cadavers." would benefit from more explanation. Are the authors talking about vascular spots/artefacts seen on DECT or artefacts on LM?

Line 97- Neurocranium as opposed to viscerocranium? Would suggest using "cranium" as it is more widely accepted.

Line 104- "iliacal vessels" spelling error

Line 106- "kidney stones"- assuming cadavers had them? As the list is supposed to be anatomic locations would change that to "kidneys".

Lines 113 and 115: would strike the word "foot" as it is already under the header "Foot".

Lines 113-114: would list joints either distal to proximal or proximal to distal for ease of reading. Additionally would replace "ankle" with "tibiotalar joint" 

Line 148- Figure 4 is presented right after Figure 1? 

Figure 1 DECT image would benefit from cropping and centering of the finding.

Figure 1 should be moved to above the "embalmed cadaver" section.

Line 159- "in the body trunk..." is redundant.

Figure 2- arrows do not point to deposits.

Line 164- would use more commonly used terms such as "costochondral cartilage" and "intervertebral disc".

Table 1 title font size is too small. Inconsistent font and font size throughout text.

Table 1 could benefit from major cleanup and format change. Hard to read.

Table 1: would use the more commonly used term "abdominal aorta" instead of "abdominalis".

why was the associated between MSU deposits and calcified plaque only tested in the ascending aorta? It seems relevant to comment whether this was tested in other vessels/parts of the aorta and what the results were.

Line 171: "all other vessels..." cannot understand this sentence. Needs to be rewritten.

Figure 3 caption: all arrows are weighted the same. Would use different types of annotations such as arrows and arrowheads.

Lines 181 and 182 need to be deleted. It seems internal reviewers had the same notion as above!

Line 184: would use "first MTP" joint instead of "MTP 1"

Table 2: same comments as table 1.

Figure 4: No (A) image but the caption mentions it. Arrows not in the right place. Weighting of arrows are inconsistent from image to image. 

-It seems polarizing microscopic evaluation was only done on fresh cadavers but then results are presented in a way that leads one to believe microscopic confirmation was performed on all specimens.

-Discussion needs to be rewritten. It does not read well.

Author Response

First of all, we would like to thank the reviewer for the detailed preparation and valuable suggestions for improvement.

Line 26- should this be greater than and equal to 5mm per classification later on in text?

It is greater than and equal, we changed it in the text, thank you very much for pointing this out!

Line 54- "To exclude artefacts, MSU crystal characterization with polarizing light microscopy was performed using the lens, aorta and feet tendons in cadavers." would benefit from more explanation. Are the authors talking about vascular spots/artefacts seen on DECT or artefacts on LM?

Thank you very much for your suggestion! We added some explanation to the introduction as following: „Nevertheless, there is still a broad discussion regarding subclinical or vascular deposits. A general consensus regarding the optimal dect protocol has not yet been es-tablished. it remains questionable how many of the msu deposits found in the dect are actual urate or merely artifacts. To our knowledge, direct imaging of MSU deposits of the head, body trunk and feet in embalmed cadavers by DECT has not been reported to date.

The role of artefacts is a frequent point of discussion. Due to the lack of verification by microscopy, we do not yet know in part, depending on the postprocessing protocol used, whether the MSU plaques are always true MSU deposits or artifacts. To exclude artefacts and verify the MSU deposits found with DECT, crystal characterization with polarizing light microscopy was performed in the lens, aorta and feet tendons in fesh cadavers.“.

Line 97- Neurocranium as opposed to viscerocranium? Would suggest using "cranium" as it is more widely accepted.

Thank you – we changed to cranium!

Line 104- "iliacal vessels" spelling error

We changed it.

Line 106- "kidney stones"- assuming cadavers had them? As the list is supposed to be anatomic locations would change that to "kidneys".

Thank you for pointing this out! We deleted „stones“.

Lines 113 and 115: would strike the word "foot" as it is already under the header "Foot".

Good point – thank you very much!

Lines 113-114: would list joints either distal to proximal or proximal to distal for ease of reading. Additionally would replace "ankle" with "tibiotalar joint"

Thank you very much, that is a good suggestion, we changed it. It is now listed from distal to proximal.

Line 148- Figure 4 is presented right after Figure 1?

Thank you for pointing this out, we overlooked that. I have changed the order of the figures.

Figure 1 DECT image would benefit from cropping and centering of the finding.

We cropped the image accordingly!

Figure 1 should be moved to above the "embalmed cadaver" section.

We moved the images!

Line 159- "in the body trunk..." is redundant.

True, we deleted it!

Figure 2- arrows do not point to deposits.

The arrows have been fixed!

Line 164- would use more commonly used terms such as "costochondral cartilage" and "intervertebral disc".

Thank you, we changed it!

Table 1 title font size is too small. Inconsistent font and font size throughout text.

Thank you for your suggestion! We changed all the tables. I hope you will find it clearer and better now.

Table 1 could benefit from major cleanup and format change. Hard to read.

Thank you for your suggestion! We changed all the tables. I hope you will find it clearer and better now.

Table 1: would use the more commonly used term "abdominal aorta" instead of "abdominalis".

We changed it.

why was the associated between MSU deposits and calcified plaque only tested in the ascending aorta? It seems relevant to comment whether this was tested in other vessels/parts of the aorta and what the results were.

It was tested in other vessels, but we could find a significant association only in the ascending aorta. We made this clear in the text: „All other vessels or regions in the body trunk demonstrated no significant association or age dependence for calcified plaques or MSU deposits.“.

Line 171: "all other vessels..." cannot understand this sentence. Needs to be rewritten.

We rewrote the sentence.

Figure 3 caption: all arrows are weighted the same. Would use different types of annotations such as arrows and arrowheads.

We changed it accordingly.

Lines 181 and 182 need to be deleted. It seems internal reviewers had the same notion as above!

Oh that's embarrassing I unfortunately overlooked that, thank you! In fact my colleagues had the same idea.

Line 184: would use "first MTP" joint instead of "MTP 1"

We changed to „First MTP“.

Table 2: same comments as table 1.

Thank you for your suggestion! We changed all the tables. I hope you will find it clearer and better now.

Figure 4: No (A) image but the caption mentions it. Arrows not in the right place. Weighting of arrows are inconsistent from image to image.

We have reformatted the images.

-It seems polarizing microscopic evaluation was only done on fresh cadavers but then results are presented in a way that leads one to believe microscopic confirmation was performed on all specimens.

We clarified it by always using the term „fresh cadavers“, whenever we wrote about polarizing microscopic evaluation.

-Discussion needs to be rewritten. It does not read well.

We rewrote the Discussion, I hope it now meets your requirements.

Reviewer 2 Report

This paper discusses the prevalence of monosodium urate (MSU) deposits in cadavers which can be detected by dual-energy computed tomography (DECT). Overall, the paper requires substantial revisions.

  1. The introduction is very short. There is no clear explanation about the problem and how it has been tackled over the past which led to the motivation in driven this research.
  2. No proper related work and recent literature is discussed in the context of analyzing MSU deposits
  3. Lots of typos. For example, “No medical history was availableincluding of gouty arthritis”
  4. The quality of all the figure are poor and pixelated
  5. Although it’s a clinical study. Still, the comparison with state-of-the-art is one of the mandatory outcomes of any research study. This is clearly missing in this manuscript.
  6. Have the authors tried utilizing deep learning to extract MSU deposits (as shown in Figure 3)? It would be interesting to know the extent of deep learning models to accurate identify MSUs. Several works have been initiated in this direction for analyzing bones, [1], tendons [2], lungs [3], and arteries [4], etc., where such deep learning systems provides great assistance to clinicians for producing accurate prognosis [5].
  7. The consent about releasing the data publicly to reproduce scientific research is missing in the manuscript.

[1] R. F. Masood et al., “Deep Learning based Vertebral Body Segmentation with Extraction of Spinal Measurements and Disorder Disease Classification”, Biomedical Signal Processing and Control, January 2022.

[2] H. Raja et al., “Clinically Verified Hybrid Deep Learning System for Retinal Ganglion Cells Aware Grading of Glaucomatous Progression”, IEEE Transactions on Biomedical Engineering, July 2021.

[3] A. M. Khan et al., “Continual Learning Objective for Analyzing Complex Knowledge Representations”, Sensors, February 2022.

[4] S. Akbar et al., “AVRDB: Annotated Dataset for Vessel Segmentation and Calculation of Arteriovenous Ratio”, 21st International Conference on Image Processing, Computer Vision & Pattern Recognition, July 17-20, 2017.

[5] M. Sirshar et al., “An Incremental Learning Approach to Automatically Recognize Pulmonary Diseases from the Multi-vendor Chest Radiographs”, Computers in Biology and Medicine, April 2021.

Author Response

First of all, we would like to thank the reviewer for the detailed preparation and valuable suggestions for improvement.

The introduction is very short. There is no clear explanation about the problem and how it has been tackled over the past which led to the motivation in driven this research.
Thank you very much for your suggestion. We have added additional explanations in the introduction. I hope we were able to emphasize the intention and question of our study and that the introduction now meets your requirements.

No proper related work and recent literature is discussed in the context of analyzing MSU deposits
We added recent literature to the paper!

(Pascart T, Carpentier P, Choi HK, Norberciak L, Ducoulombier V, Luraschi H, et al. Identification and characterization of peripheral vascular color-coded DECT lesions in gout and non-gout patients: The VASCURATE study. Semin Arthritis Rheum. 2021;51(4):895-902.

Wang P, Smith SE, Garg R, Lu F, Wohlfahrt A, Campos A, et al. Identification of monosodium urate crystal deposits in patients with asymptomatic hyperuricemia using dual-energy CT. RMD Open. 2018;4(1):e000593.

Dalbeth N, Becce F, Botson JK, Zhao L, Kumar A. Dual-energy CT assessment of rapid monosodium urate depletion and bone erosion remodelling during pegloticase plus methotrexate co-therapy. Rheumatology (Oxford). 2022.

Pascart T, Budzik JF. Dual-energy computed tomography in crystalline arthritis: knowns and unknowns. Curr Opin Rheumatol. 2022;34(2):103-10.

Tse JJ, Kondro DA, Kuczynski MT, Pauchard Y, Veljkovic A, Holdsworth DW, et al. Assessing the Sensitivity of Dual-Energy Computed Tomography 3-Material Decomposition for the Detection of Gout. Invest Radiol. 2022.

Ahn SJ, Zhang D, Levine BD, Dalbeth N, Pool B, Ranganath VK, et al. Limitations of dual-energy CT in the detection of monosodium urate deposition in dense liquid tophi and calcified tophi. Skeletal Radiol. 2021;50(8):1667-75.

Dubief B, Avril J, Pascart T, Schmitt M, Loffroy R, Maillefert JF, et al. Optimization of dual energy computed tomography post-processing to reduce lower limb artifacts in gout. Quant Imaging Med Surg. 2022;12(1):539-49.

Chou H, Chin TY, Peh WC. Dual-energy CT in gout - A review of current concepts and applications. J Med Radiat Sci. 2017;64(1):41-51.

Lots of typos. For example, “No medical history was availableincluding of gouty arthritis”
We corrected the typos. Please excuse this inattention.

The quality of all the figure are poor and pixelated
We provided bette quality images (600dpi).

Although it’s a clinical study. Still, the comparison with state-of-the-art is one of the mandatory outcomes of any research study. This is clearly missing in this manuscript.
We totally agree that comparison with state-of-the-art is necessary, but DECT is already established as a state of the art method for diagnosing and evaluating gout patients (see ACR criteria). Microscopic workup would be the highest gold standard, which has been performed in our study.

Have the authors tried utilizing deep learning to extract MSU deposits (as shown in Figure 3)? It would be interesting to know the extent of deep learning models to accurate identify MSUs. Several works have been initiated in this direction for analyzing bones, [1], tendons [2], lungs [3], and arteries [4], etc., where such deep learning systems provides great assistance to clinicians for producing accurate prognosis [5].
That is indeed a very interesting point! Since the processing of MSU plaques, especially in vessels, has not yet been well investigated and needs to be fine-tuned in terms of protocol, etc., there are unfortunately no implemented or commercially available systems based on deep learning at the moment. However, we are planning to generate a ki-based system for automated evaluation and volumetrization in further works.
The studies you presented are very interesting and useful. We have included your considerations with the citation of Masood et al., Akbar et al. and Sirshar et al., in the Discussion. Thank you very much!

The consent about releasing the data publicly to reproduce scientific research is missing in the manuscript.
Sorry, we added the statement.

Round 2

Reviewer 2 Report

The reviewers have addressed my concerns and I recommended its acceptance.